# Weighted Single-Step Genome-Wide Association Study Uncovers Known and Novel Candidate Genomic Regions for Milk Production Traits and Somatic Cell Score in Valle del Belice Dairy Sheep

**DOI:** 10.3390/ani12091155

**Published:** 2022-04-29

**Authors:** Hossein Mohammadi, Amir Hossein Khaltabadi Farahani, Mohammad Hossein Moradi, Salvatore Mastrangelo, Rosalia Di Gerlando, Maria Teresa Sardina, Maria Luisa Scatassa, Baldassare Portolano, Marco Tolone

**Affiliations:** 1Department of Animal Sciences, Faculty of Agriculture and Natural Resources, Arak University, Arak 38156-8-8349, Iran; amfarahanikh@gmail.com (A.H.K.F.); moradi.hosein@gmail.com (M.H.M.); 2Dipartimento Scienze Agrarie, Alimentari e Forestali, University of Palermo, 90128 Palermo, Italy; salvatore.mastrangelo@unipa.it (S.M.); rosalia.digerlando@unipa.it (R.D.G.); mariateresa.sardina@unipa.it (M.T.S.); baldassare.portolano@unipa.it (B.P.); marco.tolone@unipa.it (M.T.); 3Istituto Zooprofilattico Sperimentale della Sicilia “A. Mirri”, 90129 Palermo, Italy; luisa.scatassa@izssicilia.it

**Keywords:** window regions, candidate genes, dairy sheep, milk fat, milk protein, somatic cell scores

## Abstract

**Simple Summary:**

Milk production is the most economically crucial dairy sheep trait and constitutes the major genetic enhancement purpose via selective breeding. Also, mastitis is one of the most frequently encountered diseases, having a significant impact on animal welfare, milk yield, and quality. The aim of this study was to identify genomic region(s) associated with the milk production traits and somatic cell score (SCS) in Valle del Belice sheep using single-step genome-wide association (ssGWA) and genotyping data from medium density SNP panels. We identified several genomic regions (OAR1, OAR2, OAR3, OAR4, OAR6, OAR9, and OAR25) and candidate genes implicated in milk production traits and SCS. Our findings offer new insights into the genetic basis of milk production traits and SCS in dairy sheep.

**Abstract:**

The objective of this study was to uncover genomic regions explaining a substantial proportion of the genetic variance in milk production traits and somatic cell score in a Valle del Belice dairy sheep. Weighted single-step genome-wide association studies (WssGWAS) were conducted for milk yield (MY), fat yield (FY), fat percentage (FAT%), protein yield (PY), protein percentage (PROT%), and somatic cell score (SCS). In addition, our aim was also to identify candidate genes within genomic regions that explained the highest proportions of genetic variance. Overall, the full pedigree consists of 5534 animals, of which 1813 ewes had milk data (15,008 records), and 481 ewes were genotyped with a 50 K single nucleotide polymorphism (SNP) array. The effects of markers and the genomic estimated breeding values (GEBV) of the animals were obtained by five iterations of WssGBLUP. We considered the top 10 genomic regions in terms of their explained genomic variants as candidate window regions for each trait. The results showed that top ranked genomic windows (1 Mb windows) explained 3.49, 4.04, 5.37, 4.09, 3.80, and 5.24% of the genetic variances for MY, FY, FAT%, PY, PROT%, and total SCS, respectively. Among the candidate genes found, some known associations were confirmed, while several novel candidate genes were also revealed, including *PPARGC1A*, *LYPLA1*, *LEP*, and *MYH9* for MY; *CACNA1C, PTPN1, ROBO2, CHRM3,* and *ERCC6* for FY and FAT%; *PCSK5* and *ANGPT1* for PY and PROT%; and *IL26*, *IFNG*, *PEX26*, *NEGR1*, *LAP3*, and *MED28* for SCS. These findings increase our understanding of the genetic architecture of six examined traits and provide guidance for subsequent genetic improvement through genome selection.

## 1. Introduction

Milk production and udder health are economically important traits affecting dairy farming profitability. Improvement of milk production traits will directly bring greater benefits to dairy operations, and improved mastitis resistance will reduce cost of mastitis treatments [1]. Mastitis is the most common inflammatory condition that develops as a result of infection with pathogenic microorganisms or physical trauma. The expense of lost milk, as well as diminished milk output and quality, are all related to significant economic losses for the dairy sheep industry [2]. Somatic cell count (SCC) or log transformed SCC (somatic cell score, SCS) has relatively higher heritability compared to mastitis (from 0.14 to 0.19 for SCS and from 0.02 to 0.05 for mastitis) and are used as the first trait to improve mastitis resistance [3]. Uncovering regions of the genome associated with milk-related traits and SCS is essential to get inside the biological mechanism involved in their phenotypic expression [4].

In this context, genome-wide association studies (GWAS) are a good way to discover associations between single-nucleotide polymorphism (SNP) markers across the genome and a trait of interest. Nevertheless, the strength of these studies is generally conditioned by the sample available with accurate genotypic and phenotypic data. There are usually a lot of individuals with phenotypes and pedigree, but only a small number with genotypic data. Wang et al. [5] proposed the single-step GWAS (ssGWAS) as an approach to get around this limitation. The ssGWAS method assumes that all SNPs have the same variance, which is improbable in the case of characteristics with segregating main genes or quantitative trait loci (QTL). To overcome the limitations of single-step genomic BLUP (ssGBLUP), a method known as weighted single-step genomic BLUP (WssGBLUP) uses unequal variance or weights for all SNPs [6].

The WssGBLUP [6] is a method that provides the estimation of SNP effects using GEBV predicted by single-step genomic BLUP (ssGBLUP) [7] on the basis of all phenotypes, genotypes, and pedigree data. Furthermore, WssGWAS allows for unequal variances for SNPs, which increases the accuracy of SNP effect estimation [6]. Therefore, the WssGWAS might work better than classical GWAS methods when few animals are available with phenotypic and genotypic data, and the traits are under the control of a QTL with large effects [8]. As a result, using a sliding window method, WssGWAS allows researchers to give a different weight to the SNPs and to test many markers at the same time. Therefore, this approach could allow obtaining more accurate estimations of the genetic parameters than traditional GWAS, resulting in higher QTL detection power [9].

The WssGWAS has been successfully applied for milk production traits and feed efficiency in cattle [10,11,12,13] and in goats [14], and for wool and weight traits in sheep [15]. However, few studies examining milk production traits and SCS in dairy sheep have been conducted using WssGWAS.

The Valle del Belice (VDB) is the main economically important dairy breed reared in Sicily. Nowadays, part of the milk produced by this breed is transformed into two PDO (Protected Designation of Origin) cheeses: the mono-breed Vastedda della Valle del Belice and the Pecorino Siciliano. Preliminary analyses on GWAS for milk production and quality traits have been reported in the VDB breed [16]. However, no studies have been obtained so far in this breed using WssGWAS. Hence, the goals were (1) to identify SNP windows that account for the greater amount of the genetic variance in milk yield (MY), fat yield (FY) and percentage (FAT%), protein yield (PY) and percentage (PROT%), and somatic cell score (SCS) in Valle del Belice breed using a weighted single-step approach and (2) to investigate for putative candidate genes within them.

## 2. Materials and Methods

### 2.1. Phenotypes and Pedigree

The pedigree file included 5534 animals with 178 sires and 2548 dams. About 15,000 observations were collected for 1813 Valle del Belice sheep from 15 different herds between 2006 and 2016. The A4 recording procedure was followed to record the following data [17]: daily milk yield (MY), fat yield (FY), fat percentage (FAT%), protein yield (PY), protein percentage (PROT%), and milk SCC. SCC was log-transformed into somatic cell score (SCS) [18]. A detailed summary of the phenotypic records for each trait is in Table 1.

### 2.2. Genotyping and Quality Control

A total of 481 ewes were genotyped with the Illumina OvineSNP50K BeadChip. The ovine genome sequence assembly Oar_v4.0 revealed chromosomal coordinates for each SNP. Quality control was performed using the PLINK v. 1.9 [19]. Animals and markers that did not fulfil the following criteria were removed from the analysis: (i) minor allele frequency 2% or less, (ii) call rate per individual and per SNP 95% or greater and (iii) no extreme deviation from Hardy-Weinberg equilibrium (*p* ≥ 0.001, Bonferroni corrected). A total of 469 ewes and 37,228 SNPs were retained for analysis after quality control.

### 2.3. Weighted Single-Step Genome-Wide Association Study

Since this approach can use both genotyped and ungenotyped animals in the pedigree, a total of 1332 ungenotyped animals were included in the analysis. Moreover, 5534 individuals in the pedigree file were used to estimate GEBV and SNP effects, recognizing that sample size might affect the power of GWAS [8].

The analyses were carried out using the BLUPF90 family programs. In particular, the RENUMF90 package was used to renumber animals and effects sequentially and construct the input files for AIREMLF90 and BLUPF90 modules to predict GEBV [9]. Finally, the postGSf90 package was used to perform the WssGWAS [7].

The six traits were analyzed using the following single trait animal model in WssGBLUP [14]:$$y=X\beta +Z_{htd}+Z_{a}+W_{p}+e$$
where **y** is the vector of phenotypic observations, and **β** is the vector of fixed effects as reported in [16]. It was assumed that
$$\alpha \sim N(0,H\sigma_{a}^{2});\;p\sim N(0,I\sigma_{p}^{2});\;e\sim N(0,I\sigma_{e}^{2})$$
where $\sigma_{a}^{2}$, $\sigma_{p}^{2},$, and $\sigma_{e}^{2}$ were the random additive genetic variance, random permanent environmental and residual variance, respectively. **I** denoted the identity matrix, and **H** was a blend of pedigrees and the SNP derived matrix. The inverse of matrix **H** was calculated as follows:(1)$$H^{-1}=A^{-1}+\left[\begin{array}{cc} 0 & 0\\ 0 & G^{-1}-A_{22}^{-1} \end{array}\right]$$
where **A** represented the numerator relationship matrix for all individuals based on the pedigree; **A_22_** was the numerator relationship matrix for the genotyped animals; the **G** matrix was a genomic relationship matrix that was constructed according the method of VanRaden et al. [20]:$$G=\frac{ZDZ^{\prime }}{\sum_{i=1}^{M}2p_{i}\;\left(1-p_{i}\right)}$$
where **Z** is the marker matrix coded for allele frequencies (aa = 0; Aa = 1, and AA = 2), and **D** is a diagonal matrix of weights for SNP variances. **M** is the number of markers, and $p_{i}$ is equal to the minor allele frequency of the ith SNP. For ssGWAS, the SNP effects and weights were determined iteratively as follows [5]:

Step 1: in the first iteration (t = 1), **D_(t)_ = I**; **G_(t)_ = ZD_(t)_**$Z^{\prime }\lambda$, where **λ** is a normalization constant or a variance ratio that was estimated following VanRaden et al. [20]:$$\lambda =\frac{\sigma_{u}^{2}}{\sigma_{a}^{2}}=\frac{1}{\sum_{i=1}^{M}2p_{i}\;\left(1-p_{i}\right)}$$

Step 2: estimated GEBV for all animals using the ssGBLUP approach;

Step 3: calculated SNP effects as: ${\hat{u}}_{\left(t\right)}=\lambda D_{\left(t\right)}Z^{\prime }G_{\left(t\right)}^{-1}{\hat{a}}_{g}$, where ${\hat{u}}_{\left(t\right)}$ is a vector of the SNP effects estimation and ${\hat{a}}_{g}$ was the GEBV of individuals with genotypes;

Step 4: the weight of each SNP is equal to: $d_{i\left(t+1\right)}={\hat{u}}_{i\left(t\right)}^{2}2p_{i}\left(1-p_{i}\right),$ where ***i*** is the ith SNP;

Step 5: the SNP weights are normalized to keep the total genetic variance constant:$$D_{\left(t+1\right)}=\frac{tr\left(D_{\left(1\right)}\right)}{tr\left(D_{\left(t+1\right)}\right)}D_{\left(t+1\right);}$$

Step 6: the weighted matrix **G** is calculated: $G_{\left(t+1\right)}=ZD_{\left(t+1\right)}Z^{\prime }\lambda$; and

Step 7: loop back to step 2.

The number of iterations for weighting procedure in WssGWAS was determined as the iteration number that maximized the accuracy of predictions. Iterations increase the weights of SNPs with large effects and decrease those with small effects [6]. The procedure was run for five iterations on the basis of the accuracies of GEBV for our study. The proportion of variance explained by non-overlapping windows was estimated using the PostGSf90 algorithm by summing the variance of SNPs within 1 megabase (Mb) [21].

### 2.4. Detection of top SNP Windows and Functional Annotations of Candidate Genes

Putative candidate genes were identified based on initial and final coordinates of each selected window on the Oar v4.0 assembly of the ovine genome, using the NCBI Genome Data Viewer (https://www.ncbi.nlm.nih.gov/genome/gdv/?org=ovis-aries, accessed on 20 November 2015), within the top 10 windows explaining the highest percentage of additive genetic variance. The selection of the top 10 windows explaining the highest percentages of additive genetic variance is a methodology widely used in association studies to evaluate several traits in different animal species [15,22,23]. For all identified genes, literature and database searches were performed to investigate the metabolic function of the genes identified. We used the DAVID bioinformatics database (http://david.abcc.ncifcrf.gov/, accessed on 21 December 2021) to conduct Gene Ontology (GO) and Kyoto Encyclopedia of Genes and Genomes (KEGG) enrichment analysis to better understand the biological processes and pathways represented by these candidate genes. Fisher’s exact test was used to determine which terms were significantly enriched (*p* < 0.05).

## 3. Results and Discussion

### 3.1. Identification of Genomic Region and Candidate Genes

In our previous study, a classical GWAS (single-SNP GWAS) was performed for SCS in the VDB breed [16]. In the present investigation, many more genomic region windows affecting this trait were identified, and the explained genetic variance was up to 0.75%, which demonstrated that the WssGWAS was more successful in identifying window compared with our previous analysis.

Of the top 10 genomic windows identified for SCS, one was reported in our prior GWAS results and contained the gene *NEGR1* [16]. *NEGR1* (on OAR1) is a member of the LON family of immunoglobulins (IgLON) of Glycosylated (GPI-anchored) cell adhesion molecules that also includes Limbic system-associated membrane protein (LAMP), neurotrimin [24]. The gene is related with medium white blood cell count in Yak [25].

In addition, we found several genes which were not previously reported for the studied traits. For example, *ROBO2*, a fat-related QTL in marbling score [26], was identified as a novel candidate gene for milk fat yield and protein yield in the present study.

Across the six traits, the total genetic variation explained by the top 10 ranked windows varied from 3.49% (MY) to 5.24% (SCS). The largest genetic variance explained by a window was 0.75% for SCS and 0.64% for milk fat percentage, respectively. However, the majority of the windows explained less than 0.5%, and these low-contributing regions spread throughout the genome for all analyzed traits. This suggests that both milk production traits and SCS are moderately to highly polygenic, that is many regions over the genome contribute to the genetic variation of the traits (Figure 1).

The top 10 ranked windows for each trait by that explained the largest proportion of additive genetic variance and the important candidate genes for all investigated traits are presented in Table 2, Table 3, Table 4 and Table 5. The characteristics of these window regions for the six traits are explained in detail in the following paragraphs.

### 3.2. Candidate Genes for Milk Yield (MY)

Using the WssGWAS method, the top 10 identified genomic regions related with milk yield, in terms of percentage of genetic variance explained (%), were detected on chromosomes 3, 4, 6, 9, 22, and 27. The discovered genomic regions explained 0.29–0.39% of the additive genetic variances and jointly explained 3.49% for MY. Within these windows, there were 33 known genes (Figure 1A and Table 2).

The most important window region was located on OAR6: 47,499,788–48,496,456 bp, and explained 0.33% of the genetic variance for MY.

Two genes were previously reported to directly regulate milk yield [27,28]. The *PPARGC1A* gene is involved in the activation of various significant hormone receptors, in thermogenesis, gluconeogenesis, glucose transport, and β-oxidation of fatty acids [29].

Polymorphism of *PPARGC1A* gene in Italian Mediterranean buffaloes was associated with milk yield and protein percentage [28]. In Italian Holstein cattle, the SNPs harbored in *PPARGC1A* gene on BTA6 were linked with milk yield, protein yield and percentage [27]. Consequently, *PPARGC1A* could act as a marker gene for milk production traits also in the Valle del Belìce breed. *LYPLA1* has been reported as candidate gene in the regulation of prolactin secretion, feed intake and gain in cattle [30], and as marker involved in milk yield in Chios dairy sheep population [31].

The second most important window (OAR4: 92,470,191–93,383,584 bp) was located inside the leptin (*LEP*) gene, which is of great importance in animal growth and metabolism [32]. It controls feed intake, energy metabolism, and body fat distribution [33]. Genetic variants identified in *LEP* have been proven to affect milk production in cows [34]. Furthermore, a polymorphism located in the *LEP* gene was found to be associated with milk production in Najdi sheep breed [35].

Other genes were demonstrated to have mechanisms indirectly related to milk yield [36]. The *MYH9* gene is a member of the myosin superfamily which shares the common characteristics of the hydrolytic activity of ATPase, actin binding, and potential for kinetic energy transduction. Myosin plays an important role in muscle growth and contraction [37]. Moreover, Lopdell et al. [38] found that *MYH9* is associated with milk phenotypes in cattle together with *CSF2RB*.

### 3.3. Candidate Genes for Milk Fat Yield (FY) and Fat Percentage (FAT%)

For FY, there were nine chromosomes that contain associated genomic regions, including chromosomes OAR1, 2, 3, 5, 6, 7, 11, 13, and 19. The percentage of genetic variance for these windows ranged from 0.30 to 0.55%. Within these regions, 56 known genes were identified to be associated with FY (Figure 1B and Table 3).

Two candidate genes were previously reported for FY: *CACNA1C* [39] and *PTPN1* [40]. *CACNA1C* is part of a group of genes that give instructions for constructing calcium channels. Long-chain fatty acids are implicated in the calcium channel activation processes, probably acting at some near lipid binding sites on these channels or directly over the channel protein itself [39]. The *CACNA1C* was reported as a milk fatty acids profile candidate gene in Santa Inês ewes [39].

The gene *PTPN1* is associated with milk cholesterol content. In fact, it is a key gene for plasma total and high-density lipoprotein cholesterol (HDL-CHL) [40] and directly linked to CHL. A region on OAR1 in position 156,102,645–157,082,096 bp was found to be associated with milk fat and protein yield. This region contains the gene *ROBO2*. This window was detected within guidance receptor 2 (ROBO2) of the ROBO family [41]. According to previous studies, *ROBO2* was associated with fat metabolism, particularly in the fatty acid profile and includes C18:3 IMF [26,42].

For FAT%, top 10 windows in seven different chromosomes (OAR2, 6, 7, 8, 9, 25, and 26) were identified (Figure 1D and Table 3). Results showed that these windows explained 0.41–0.64% of the genetic variance for FAT%.

The most important window, OAR25: 12,322,443–13,278,443 bp, accounted for 0.48% of the genetic variance of FAT% and was found within the cholinergic receptor gene, muscarinic 3 (*CHRM3*). This gene plays an essential role in lipid metabolism, which may have an effect on the milk fat yield [43]. Mei et al. [44] stated that highly significant SNPs for meat quality in Qinchuan cattle were found inside *CHRM3* gene.

Another important window for fat yield was found on chromosome 25 between 45,303,771 and 46,293,771 bp. This region contained *ERCC6*, an important candidate gene involved in milk fat in Canadian Holsteins [45]. The involvement of *ERCC6* gene in milk production has been poorly investigated so further studies are needed.

### 3.4. Candidate Genes for Milk Protein Yield and Protein Percentage

Top 10 windows genomic regions with a high impact on PY were located on chromosomes 1, 2, 3, 7, 8, 9, 11, 13, and 14. The proportion of genetic variance for these windows ranged from 0.30 to 0.51%. A total of 44 known genes associated with protein yield were identified in these regions (Figure 1C and Table 4).

From these regions, genomic regions located on chromosome 2 (from 63,313,028 to 64,303,381 bp) were found in protein yield. *PCSK5* gene product is known to directly inactivate endothelial lipase and indirectly cleave and activate angiopoetin-like protein 3, a natural inhibitor of endothelial lipase. A recent study reports *PCSK5* as gene linked to meat quality traits in beef cattle [46]. Further studies are necessary to clarify the role of the *PCSK5* gene in the milk protein yield association.

Relevant window for protein yield was found on OAR9 between 77,706,323 and 78,625,358 bp. Among the putative candidate genes mapped within this window, we found Angiopoietin-1 (*ANGPT1*), which explained more than 0.35% of the genetic variance. *ANGPT1* is mainly produced by cardiac, skeletal and smooth muscle cells, and adventitial cells [47]. The association of *ANGPT1* gene with milk quality in Chinese Holstein has been reported in a previous study [48].

Regarding milk protein percentage, the top window identified OAR1, 2, 3, 5, 7, 13, and 17, together with 30 genes close to the most important SNP within each window (Figure 1E and Table 4). The identified genomic windows explained 0.33–0.50% of the genetic variance for PROT%. Interestingly, these genomic regions were not reported in recognized QTL regions for milk protein percentage.

### 3.5. Candidate Genes for SCS

Genomic windows linked to SCS were found in five chromosomes, OAR1, 3, 6, 7, 9 where 37 known genes were associated with the trait (Figure 1F and Table 5). The proportion of genetic variance for these windows ranged from 0.41 to 0.75%.

The most important windows, OAR3: 161,701,025–162,705,420 bp, and 213,347,420–214,351,547 bp, contributed to 0.90% of the genetic variance of SCS.

Three genes were previously reported as candidate for immune response, namely, *IL26* [49], *IFNG* [50], and *PEX26* [51].

The protein encoded by *IL26* is a member of the IL10 family of cytokines [52], which has anti-bacterial activity against various bacteria, including *Staphylococcus aureus* [49], suggesting a broader role in host defenses.

The critical function of the *IFNG* gene in the coordination of immunity against bacteria suggested a potential role of this gene in the bovine paratuberculosis susceptibility [50]. In addition, the *IFNG* gene was previously associated with host innate immune response against Brucella in goats [53].

*PEX26* is a peroxisomal membrane protein that acts as a membrane anchor for the PEX1-PEX6 complex and has been previously reported to be associated with immune response traits in Canadian Holstein cattle [51].

Additionally, other genes were demonstrated to have mechanisms related to SCS. Previous significant QTL on chromosome 6 containing candidate genes (*LAP3*, *MED28*, *FAM184B*, *DCAF16*, *NCAPG*, and *LCORL*) related to milk production traits in cattle [1,13], were also identified in this study. In this region, *MED28* could be a causative gene affecting SCS in Valle del Belice sheep. The gene *MED28* is expressed in the mammary gland during lactation and is associated to breast cancer [54], whereas *LAP3* is a regulator of hormone and plays a role in maturation, inactivation, and degradation of proteins [55]. In mammals, LAPs contribute to processing of some bioactive peptides and vesicle trafficking to the plasma membrane, and they also have a role in MHC I antigen presentation [56].

### 3.6. Functional Annotation of Enrichment Analysis

In order to better understand the biological processes and pathways, genes surrounding the SNPs within each window were searched. Hence, KEGG and GO enrichment analysis were performed. Five GO terms and one KEGG pathways were enriched for the analyzed milk production traits and somatic cell score. The enriched GO terms are involved in positive regulation of protein phosphorylation (GO: 0001934) consisting of *ANGPT1* and *PTPN1* genes; positive regulation of protein complex assembly (GO: 0031334) consisting of *ERCC6*, *RAP1B*, and *LCP1* genes; lipoprotein metabolic process (GO: 0042157) including *LYPLA1* and *SCARB1* genes; regulation of leukocyte proliferation (GO: 0002682) including *DLG1*, *IL7*, *LAPTM5*, and *TFRC* genes; regulation of immune system process (GO: 0031334) consisting of *IFNG*, *IL7*, and *LAPTM5* genes; and IL-17 signaling pathway (oar: 04657) consisting of *IFNG*, *IL17RB*, and *MMP9*. The results of GO and KEGG enrichment analyses further suggests that milk production and somatic cell score are traits in which many genes are involved.

## 4. Conclusions

This research conducted a weighted GWAS using a single-step procedure and found many genomic regions and candidate genes in relation to five milk production traits and SCS, indicating a polygenic nature of the studied traits. Novel candidate genes implicated in milk production traits and SCS were also identified. Our findings offer new insights into the genetic basis of milk production traits and SCS in Valle del Belice sheep, and more in general in dairy sheep. These results can be used to search for causative mutations and for breeding through marker-assisted selection to improve the production and quality of milk yield in Valle del Belice sheep. Further research with more information on animals, records, and genotypes is required to validate our findings and the practical implications of this for genomic selection.

## Figures and Tables

**Figure 1 animals-12-01155-f001:**
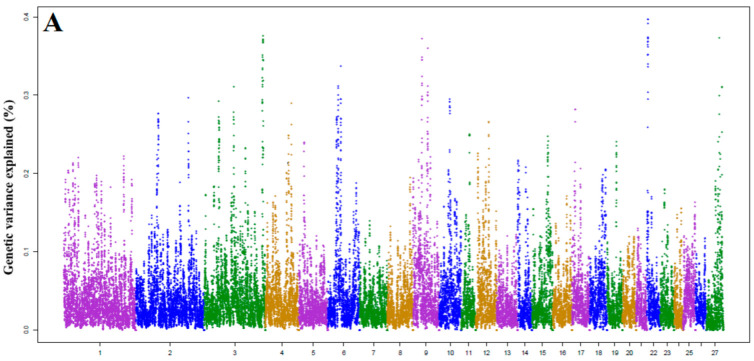
Manhattan plots of the additive genetic variance (%) explained by 1.0 Mb window of adjacent SNPs for milk production traits and SCS: (**A**) milk yield, (**B**) fat yield, (**C**) protein yield, (**D**) fat percentage, (**E**) protein percentage, and (**F**) SCS. Each dot represents a window.

**Table 1 animals-12-01155-t001:** Descriptive statistics for studied traits.

Traits	Number	Mean ± SD	CV(%)	Min-Max	h^2^
MY (g)	15,008	1318 ± 552	41.91	62–4140	0.10
FY (g)	15,008	91.06 ± 34.91	38.34	3.91–393.53	0.06
FAT (%)	15,008	7.08 ± 1.09	15.41	2.53–10.80	0.11
PY (g)	15,008	75.43 ± 29.81	39.52	2.93–238.98	0.09
PROT (%)	15,008	5.80 ± 0.65	11.16	2.14–8.10	0.15
SCS	15,008	2.67 ± 0.72	0.27	1–5.31	0.04

SD, standard deviation; CV, coefficient of variation; Min-Max, minimum and maximum values; h^2^: heritability.

**Table 2 animals-12-01155-t002:** Identification of genes based on additive genetic variance explained by top 10 window regions for milk yield trait.

Trait	OAR	Start (bp)	Stop (bp)	% VE	Genes
MY	22	5,250,187	6,225,066	0.396	*PCDH15, U6*
	3	214,344,054	215,239,751	0.375	*EPS8*, *PTPRO*, *RERG*, *PDE6H*, *ARHGDIB*, *ERP27*, *MGP*
	27	106,515,599	107,465,396	0.373	*-*
	9	31,897,584	32,871,881	0.372	*C9H8orf76*, *TBC1D31*, *DERL1*, *ZHX2*
	9	54,661,246	55,632,383	0.359	*TMEM70*, *LY96*, *JPH1*, *GDAP1*, *MIR2052*
	3	212,632,779	213,605,435	0.351	*LMO3*, *MGST1*
	6	47,499,788	48,496,456	0.337	*PPARGC1A*
	6	40,445,167	41,445,859	0.311	*LYPLA1*, *FAM13A*, *NAP1L5*
	3	107,748,212	108,707,604	0.310	*LYG2*, *MRPL30*, *MITD1*, *C2orf15*, *MYH9*, *TSGA10*, *MGAT4A*
	4	92,470,191	93,383,584	0.291	*KCND2*, *LEP*, *TSPAN12*

OAR = *ovis aries* chromosome; %VE = percentage of explained genetic variance.

**Table 3 animals-12-01155-t003:** Identification of genes based on additive genetic variance explained by top 10 window regions for milk fat yield and fat percentage traits.

Trait	Chr	Start (bp)	Stop (bp)	% VE	Genes
FY	11	17,769,576	18,742,505	0.551	*GOSR2*, *RPRML*, *LYZL6*, *RDM1*, *PLEKHM1*, *ARHGAP27*, *MAP3K14*, *FMNL1*, *HEXIM2*, *HEXIM1*, *ACBD4*, *PLCD3*, *NMT1*
	3	151,003,018	151,972,475	0.483	*ANO6*, *DBX2*, *NELL2*
	6	3,644,480	4,610,354	0.420	*BBS7*, *CCNA2*, *EXOSC9*, *SMIM43*
	19	5,602,339	6,570,519	0.404	*GADL1*, *OSBPL10*, *STT3B*
	3	229,350,131	230,350,131	0.376	*ATP6V1E1*, *TUBA8*, *CDC42EP1*, *LGALS2*, *GGA1*, *CACNA1C*, *U6*, *PDXP*
	2	56,504,189	57,464,796	0.373	*-*
	7	37,844,731	38,839,931	0.373	*SNAP23*, *LRRC57*, *HAUS2*, *CDAN1*, *TTBK2*, *UBR1*, *TMEM62*, *CCNDBP1*, *EPB42*
	5	47,771,920	48,762,985	0.368	*TCF7*, *CDKL3*, *UBE2B*, *JADE2*, *SAR1B*, *SEC24A*, *CAMLG*, *DDX46*, *C5orf24*
	1	156,102,645	157,082,096	0.357	*ROBO2*, *U2*
	13	81,534,944	82,519,944	0.301	*UBE2V1*, *CEBPB*, *PTPN1*, *PARD6B*, *BCAS4*, *DPM1*, *KCNG1*, *MOCS3*
FAT%	6	3,563,877	4,559,472	0.649	*NAF1*, *BBS7*, *CCNA2*, *EXOSC9*, *SMIM43*
	26	3,879,792	4,875,104	0.619	*-*
	8	82,898,823	83,853,076	0.571	*ESR1*, *SYNE1*, *MYCT1*, *VIP*
	8	79,925,767	80,905,873	0.550	*UST*, *TAB2*, *ZC3H12D*, *PPIL4*, *GINM1*, *KATNA1*, *LATS1*, *NUP43*, *PCMT1*, *LRP11*
	26	700,916	1,677,676	0.540	*DLGAP2*, *CLN8*, *CETN2*, *KBTBD11*, *MYOM2*
	9	53,991,923	54,979,070	0.533	*RDH10*, *ELOC*, *TMEM70*, *LY96*
	2	124,285,685	125,280,477	0.527	*FAM168B*, *PLEKHB2*
	25	12,322,443	13,278,443	0.483	*ZNF248*, *BMS1*, *CHRM3*, *ZNF33B*
	7	50,459,248	51,446,200	0.470	*RORA*, *ICE2*, *ANXA2*
	25	45,303,771	46,293,771	0.419	*DRGX*, *ERCC6*, *SLC18A3*, *C25H10orf53*, *OGDHL*, *PARG*, *TIMM23B*, *SNORA74*, *MARCHF8*, *ZFAND4*

**Table 4 animals-12-01155-t004:** Identification of genes based on additive genetic variance explained by top 10 window regions for milk protein yield and protein percentage traits.

Trait	Chr	Start (bp)	Stop (bp)	% VE	Genes
PY	7	37,844,731	38,839,931	0.518	*SNAP23*, *LRRC57*, *HAUS2*, *CDAN1*, *TTBK2*, *UBR1*, *TMEM62*, *CCNDBP1*
	8	14,481,631	15,471,954	0.505	*HDDC2*, *TPD52L1*, *RNF217*, *NKAIN2*
	13	12,488,726	13,487,350	0.449	*SFMBT2*, *ITIH5*, *ITIH2*, *ATP5F1C*, *TAF3*, *GATA3*
	2	111,987,606	112,986,964	0.398	*KIF13B*, *MSRA*, *PRSS51*, *PRSS55*, *PINX1*
	14	25,500,650	26,497,592	0.397	*MT1A*, *MT1C*, *MT2*, *OGFOD1*, *OGFOD1*, *NUDT21*, *AMFR*, *GNAO1*, *CES5A*
	2	63,313,028	64,303,381	0.393	*VPS13A*, *FOXB2*, *GCNT1*, *RFK*, *PCSK5*
	2	59,313,028	60,303,381	0.388	*-*
	1	156,123,514	157,114,945	0.379	*ROBO2*, *U2*
	9	77,706,323	78,625,358	0.354	*ANGPT1*, *ABRA*, *OXR1*
	3	151,003,018	151,972,475	0.305	*ANO6*, *DBX2*, *NELL2*
PROT%	13	77,917,677	78,910,066	0.507	*DNTTIP1*, *TNNC2*, *SNX21*, *ACOT8*, *CTSA*, *PLTP*, *PCIF1*, *ZNF335*, *MMP9*
	2	80,632,041	81,604,796	0.452	*KDM4C*, *DMAC1*
	17	29,494,464	30,441,072	0.420	*-*
	1	38,670,059	39,649,763	0.368	*TM2D1*, *PATJ*, *KANK4*
	7	37,916,711	38,872,804	0.357	*CDAN1*, *TTBK2*, *UBR1*, *TMEM62*, *EPB42*
	3	179,367,669	180,265,170	0.357	*-*
	5	72,168,942	73,159,451	0.342	*SOX30*, *THG1L*, *LSM11*, *CLINT1*
	2	134,606,095	135,598,790	0.337	*TTC21B*, *GALNT3*
	2	154,466,090	155,460,561	0.332	*ZNF804A*
	13	76,376,357	77,356,228	0.332	*YWHAB*, *PABPC1L*, *STK4*, *KCNS1*, *MATN4*

**Table 5 animals-12-01155-t005:** Identification of genes based on additive genetic variance explained by top 10 window regions for SCS.

Trait	Chr	Start (bp)	Stop (bp)	% VE	Genes
SCS	1	33,955,437	34,931,948	0.758	*U6*, *OMA1*, *TACSTD2*
	6	41,728,563	42,727,105	0.677	*LAP3*, *MED28*, *FAM184B*, *DCAF16*, *NCAPG*, *LCORL*
	9	34,510,191	35,480,917	0.562	*PRKDC*, *MCM4*, *EFCAB1*, *SNAI2*, *PPDPFL*
	1	32,365,838	33,350,564	0.557	*PLPP3*, *PRKAA2*, *C8A*, *DAB1*
	7	38,236,462	39,205,426	0.485	*UBR1*, *TMEM62*, *CCNDBP1*, *EPB42*
	3	213,347,420	214,351,547	0.484	*LMO3*, *MGST1*, *SLC15A5*, *PEX26*, *STRAP*
	1	49,775,153	50,755,499	0.463	*NEGR1*
	3	161,701,025	162,705,420	0.421	*CPM*, *SLC35E3*, *NUP107*, *RAP1B*, *MDM1*, *IL22*, *IL26*, *IFNG*
	9	28,511,250	29,492,867	0.420	*LRATD2*
	1	61,734,830	62,648,480	0.413	*ADGRL2*

## Data Availability

https://osf.io/ajwcb/?view_only=92ab35dec6564d28bcc259c549509a0b Accessed Date: 28 November 2021.

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
