# Peer review of "Weighted Single-Step Genome-Wide Association Study Uncovers Known and Novel Candidate Genomic Regions for Milk Production Traits and Somatic Cell Score in Valle del Belice Dairy Sheep"

_animals, 2022, doi:10.3390/ani12091155_

Round 1
Reviewer 1 Report
In my opinion, the study was well conducted, as well the manuscript written.
I have only two considerations:
Some of the terms presented in the keywords coincide with the title and abstract (eg Weighted single-step GWAS). I suggest replacing them.
Figures D, E and F were not shown, compromising the interpretation of the results. Please add figures D, E and F.
Author Response
We would like to thank you for giving us the opportunity to revise and improve our manuscript. We are re-submitting a new version of the manuscript to be taken into account for publication in your valuable journal. Thank you very much for your kind consideration.
Reviewer #1:
In my opinion, the study was well conducted, as well the manuscript written. I have only two considerations:
Answer: We would like to thank the reviewer for the positive feedback. We appreciate the comment. The revisions are carried out in the new version.
Point 1: Some of the terms presented in the keywords coincide with the title and abstract (eg Weighted single-step GWAS). I suggest replacing them.
Response 1: Thank you for the comment. Corrected in the revised version.
Point 2: Figures D, E and F were not shown, compromising the interpretation of the results. Please add figures D, E and F.
Response 2: Thank you for the comment. The Figures D, E and F are added in the revised version.

Reviewer 2 Report
Minor revison:
In line 126, page 3,
there may be some typos in this forumal.
Author Response
We would like to thank you for giving us the opportunity to revise and improve our manuscript. We are re-submitting a new version of the manuscript to be taken into account for publication in your valuable journal. Thank you very much for your kind consideration.
Reviewer #2:
Comments and Suggestions for Authors
Minor revision:
Answer: We would like to thank the reviewer for the positive feedback. We appreciate the comment. The revisions are carried out in the new version.
Point 1: In line 126, page 3, there may be some typos in this forumal.
Response 1: Thank you for the comment. This formula is related to our previous article in 2021 in Frontiers in Genetics journal.
Sutera et al., 2021. Genome-Wide Association Study Identifies New Candidate Markers for Somatic Cells Score in a Local Dairy Sheep. Front. Genet. 12:643531.

Reviewer 3 Report
The study is interesting providing new QTLs affecting milk traits in sheep. However, the authors should discuss why they do not establish a threshold for selecting windows. For example, SNP windows that explained 1% or more of the total genetic variance based on the WssGWAS have been selected on the literature:
- Marques, D.B.D., Bastiaansen, J.W.M., Broekhuijse, M.L.W.J. et al. Weighted single-step GWAS and gene network analysis reveal new candidate genes for semen traits in pigs. Genet Sel Evol 50, 40 (2018). https://doi.org/10.1186/s12711-018-0412-z
- Lemos MV, Chiaia HL, Berton MP, Feitosa FL, Aboujaoud C, Camargo GM, et al. Genome-wide association between single nucleotide polymorphisms with beef fatty acid profile in Nellore cattle using the single step procedure. BMC Genomics. 2016;17:213.
- Irano N, de Camargo GM, Costa RB, Terakado AP, Magalhaes AF, Silva RM, et al. Genome-wide association study for indicator traits of sexual precocity in Nellore cattle. PLoS One. 2016;11:e0159502.
- Gonzalez-Pena D, Gao G, Baranski M, Moen T, Cleveland BM, Kenney PB, et al. Genome-wide association study for identifying loci that affect fillet yield, carcass, and body weight traits in rainbow trout (Onchorhynchus mykiss). Front Genet. 2016;7:203.
I understand that the proportion of variance found is lower, but authors should justify and discuss this point.
Another point that could enrich the manuscript is to perform a network analysis, or an enrichment functional annotation analysis using softwares as DAVID, GeneCodis or Cytoscape.
Some sentences are very similar or equal to those in the manuscript by Yin et al., 2019 ( https://doi.org/10.3389/fgene.2019.01053). For example lines 188 to 193 are very similar to the first paragraph of the Discussion section of the manuscript by Yin et al., 2019. Please, authors should rewrite those paragraphs. The same in Material and methods.
Author Response
We would like to thank you for giving us the opportunity to revise and improve our manuscript. We are re-submitting a new version of the manuscript to be taken into account for publication in your valuable journal. Thank you very much for your kind consideration.
Reviewer #3:
Answer: We would like to thank the reviewer for the positive feedback. We appreciate the comment. The revisions are carried out in the new version.
Comments and Suggestions for Authors
Point 1: The study is interesting providing new QTLs affecting milk traits in sheep. However, the authors should discuss why they do not establish a threshold for selecting windows. For example, SNP windows that explained 1% or more of the total genetic variance based on the WssGWAS have been selected on the literature:
- Marques, D.B.D., Bastiaansen, J.W.M., Broekhuijse, M.L.W.J. et al. Weighted single-step GWAS and gene network analysis reveal new candidate genes for semen traits in pigs. Genet Sel Evol 50, 40 (2018). https://doi.org/10.1186/s12711-018-0412-z
- Lemos MV, Chiaia HL, Berton MP, Feitosa FL, Aboujaoud C, Camargo GM, et al. Genome-wide association between single nucleotide polymorphisms with beef fatty acid profile in Nellore cattle using the single step procedure. BMC Genomics. 2016;17:213.
- Irano N, de Camargo GM, Costa RB, Terakado AP, Magalhaes AF, Silva RM, et al. Genome-wide association study for indicator traits of sexual precocity in Nellore cattle. PLoS One. 2016;11:e0159502.
- Gonzalez-Pena D, Gao G, Baranski M, Moen T, Cleveland BM, Kenney PB, et al. Genome-wide association study for identifying loci that affect fillet yield, carcass, and body weight traits in rainbow trout (Onchorhynchus mykiss). Front Genet. 2016;7:203.
I understand that the proportion of variance found is lower, but authors should justify and discuss this point.
Response 1: Thank you for the comment. A detailed description has been included.
The selection of the top 10 windows explaining the highest percentages of additive genetic variance is a methodology widely used in association studies to evaluate several traits in different animal species (Tiezzi et al., 2015; Irano et al., 2016; Sevillano et al., 2019; Otto et al., 2020; Zhao et al., 2021).
Also included in other studies, e.g. Atashi et al. (2020) considered 165 Kb SNP windows that explained more than 0.20% of the genetic variance as the threshold to declare significance. Raschia, et al. (2020) using the cattle SNP50 v2 BeadChip, considered 10- consecutive SNP windows that explained more than 0.31% of total genetic variance as the threshold to declare significance. de Oliveira Silva, et al. (2017) using the BovineHD SNP panel, considered 50-adjacent SNP windows (with average of 280 kb) that explained more than 0.50% of additive genetic variance as the threshold to declare significance. In our study, the lowest percentages of additive genetic variance within top ten was 0.291.
Point 2: Another point that could enrich the manuscript is to perform a network analysis, or an enrichment functional annotation analysis using softwares as DAVID, GeneCodis or Cytoscape.
Response 2: Thank you for the comment. Enrichment functional annotation analysis using DAIVD software are added in the revised version including Materials and Methods and results section.
Methods:
In order to better understand the biological processes and pathways shared by these candidate genes, we conducted GO and KEGG enrichment analysis using DAVID bioinformatics resource (http://david.abcc.ncifcrf.gov/accessed on 21 December 2021). Significantly enriched terms were assessed using Fisher’s exact test (p < 0.05).
Results
In an attempt to better understand the biological processes and pathways shared by the trait-associated genes, we searched genes near the SNPs within each window of the six milk production traits and somatic cell score. We then performed KEGG and GO enrichment analysis. Five GO terms and one KEGG pathways were enriched for the milk production traits and somatic cell score analyzed. The enriched GO terms are involved in positive regulation of protein phosphorylation (GO: 0001934) consisting of ANGPT1 and PTPN1 genes; positive regulation of protein complex assembly (GO: 0031334) consisting of ERCC6, RAP1B, and LCP1 genes; lipoprotein metabolic process (GO: 0042157) including LYPLA1 and SCARB1 genes; regulation of leukocyte proliferation (GO: 0002682) including DLG1, IL7, LAPTM5, and TFRC genes; regulation of immune system process (GO: 0031334) consisting of IFNG, IL7, and LAPTM5 genes; and IL-17 signaling pathway (oar: 04657) consisting of IFNG, IL17RB, and MMP9. Results in GO and KEGG enrichment analyses further extend to suggest that many genes are involved with milk production and somatic cell score.
Point 3: Some sentences are very similar or equal to those in the manuscript by Yin et al., 2019 ( https://doi.org/10.3389/fgene.2019.01053). For example lines 188 to 193 are very similar to the first paragraph of the Discussion section of the manuscript by Yin et al., 2019. Please, authors should rewrite those paragraphs. The same in Material and methods.
Response 3: Thank you for the comment. Corrected in the revised version.

Reviewer 4 Report
Line 53: Change sheep dairy to dairy sheep
Line 54: Please add the h2 values
Lines 97-104: Last line should be the first of the paragraph. Average relatedness and pedigree completeness would be interesting.
Table 1: CV% for MY, FY and PY are quite high. It is adviseable to explain its reason in the results
Lines 186-201: Sounds like Materials and Methods. Should be inserted to the beginning of that chapter.
Lines 202 to 212: Variance components standard errors and heritability estimations are missing from the results
Figure 1: Please add larger font size to the y axis
Author Response
We would like to thank you for giving us the opportunity to revise and improve our manuscript. We are re-submitting a new version of the manuscript to be taken into account for publication in your valuable journal. Thank you very much for your kind consideration.
Reviewer #4:
Answer: We would like to thank the reviewer for the positive feedback. We appreciate the comment. The revisions are carried out in the new version.
Comments and Suggestions for Authors
Point 1: Line 53: Change sheep dairy to dairy sheep
Response 1: Thank you for the comment. Corrected in the revised version.
Point 2: Line 54: Please add the h2 values
Response 2: Thank you for the comment. The h2 related to traits is added in the revised version (Table 1).
Point 3: Lines 97-104: Last line should be the first of the paragraph. Average relatedness and pedigree completeness would be interesting.
Response 3: Thank you for the comment. Corrected in the revised version.
Point 4: Table 1: CV% for MY, FY and PY are quite high. It is advisable to explain its reason in the results
Response 4: Thank you for the comment. The CV for milk, fat and protein yields, and SCS were between 27% and 42% and were in line with the coefficients of variation found in other studies (Sutera et al., 2021a; Sutera et al., 2021b).
Sutera, A. M., Tolone, M., Mastrangelo, S., Di Gerlando, R., Sardina, M. T., Portolano, B., ... & Riggio, V. (2021). Detection of genomic regions underlying milk production traits in Valle del Belice dairy sheep using regional heritability mapping. Journal of Animal Breeding and Genetics, 138(5), 552-561.
Sutera, A. M., Moscarelli, A., Mastrangelo, S., Sardina, M. T., Di Gerlando, R., Portolano, B., & Tolone, M. (2021). Genome-wide association study identifies new candidate markers for somatic cells score in a local dairy sheep. Frontiers in Genetics, 12, 409.
Point 5: Lines 186-201: Sounds like Materials and Methods. Should be inserted to the beginning of that chapter.
Response 5: Thank you for the comment. Corrected in the revised version.
Point 6: Lines 202 to 212: Variance components standard errors and heritability estimations are missing from the results
Response 6: Thank you for the comment. The information related to heritability are added in the revised version (Table 1). More details on both Variance components standard errors and heritability are provided in a previous study conducted by the same authors (Sutera et al. 2021).
Sutera, A. M., Tolone, M., Mastrangelo, S., Di Gerlando, R., Sardina, M. T., Portolano, B., ... & Riggio, V. (2021). Detection of genomic regions underlying milk production traits in Valle del Belice dairy sheep using regional heritability mapping. Journal of Animal Breeding and Genetics, 138(5), 552-561.
Point 7: Figure 1: Please add larger font size to the y axis
Response 7: Thank you for the comment. Corrected in the revised version.

Round 2
Reviewer 3 Report
The manuscript has improved a lot after revision, and it is suitable for publication.